

# Characteristics of high monsoon wind-waves observed at multiple stations in the eastern Arabian Sea

M. M. Amrutha and V. Sanil Kumar

Ocean Engineering Division, CSIR-National Institute of Oceanography, Dona Paula 403 004, Goa, India

*Correspondence to*: Sanil Kumar (sanil@nio.org)

**Abstract.**   The growth and decay of surface wind-waves during one-month period in a typical Indian summer monsoon is investigated based on the data collected at 9 to 15 m water depth at 4 locations in the nearshore waters of the eastern Arabian Sea covering a spatial distance of ~350 km. The significant wave height varied from 0.7 to 5.5 m during the data collection

considered in the analysis. The heights of waves during the measurement period often exceed 3 m. The most extreme wave height is 1.50 to 1.62 times the significant wave height and the most extreme crest height of the wave is 1.23 to 1.35 times the significant wave height of the same 30-minutes record. The average ratio of crest height of the wave to the height of the same wave is 0.58 to 0.67. The height of waves having maximum crest height is smaller than the maximum wave height during 30 minutes period. Measured waves are predominantly swell, but since the majority of wave generation during the

monsoon is adjacent to the study area and the wind–wave coupling is strong, wave periods are rarely above 15 s. The numerical wave model could estimate the wave height reasonably well during the wave growth compared to the wave decay period. Hovmöller diagrams show a considerable spatial variability in the wave and wind pattern in the Indian Ocean during the high wave event at the eastern Arabian Sea.

## 1 Introduction

The characteristics of the wind generated surface waves in the eastern Arabian Sea changes drastically during the Indian summer monsoon (June to September) compared to rest of the year (Sanil Kumar et al., 2014). Due to the atmospheric circulation and topographical effects, the monsoon generates severest sea states in the Arabian Sea (AS) with waves propagating from the southwest. The characteristics of the monsoon waves in the eastern AS are described by many researchers (Rao and Baba, 1996; Sanil Kumar et al., 2000; Amrutha et al. 2015). The influence of monsoon variability on

the surface waves using measured wave data covering seven years and ERA-Interim reanalysis data (Dee et al., 2011) from 1979 to 2015 in the eastern AS is reported by Sanil Kumar and Jesbin George (2016). The occurrence of monsoon waves higher than 4 m once or more times every year in the eastern AS has become common in recent years (Sanil Kumar and Jesbin George, 2016; Anjali Nair and Sanil Kumar, 2017). Monsoon events are an important driver of coastal erosion (Komar, 1997) and cause extensive beach erosion along the west coast of India, whereas the post-monsoon long-period



waves are responsible for accretion and shaping the long-term planform geometry of the beaches (Sharma et al., 2017). The wave generation during the high monsoon winds is complex due to the interaction of atmospheric fields with oceanographic processes. Hence, there is a large interest in understanding the characteristics of high monsoon waves in the shallow water and its spatial variability along the coast. Amrutha et al. (2015) studied the wave characteristics during and after the onset of monsoon and found that just after the onset of monsoon, the wave spectra change from bi-modal to uni-modal.

The earlier studies have shown that the Rayleigh distribution over-predicts the probability of occurrence of large waves compared with field data (Massel, 1996), whereas Mori et al. (2002) and Stansell (2004) observed that the Rayleigh distribution under-predict the probability of occurrence of freak waves. Most of the earlier studies in the eastern AS describe the variations in the bulk wave parameters obtained from the wave spectrum and do not describe the crest and trough heights of high waves. For practical applications, it is important to know the ratio of the crest height to wave height of an individual wave in oceans at a given place. The crest height during the extreme waves forms the crucial parameter for designing the air gap of the marine fixed structures and assessing the impact of wave slamming force on the deck structures. A number of probability distributions are proposed for estimating the extreme crest height (Stansell, 2004). However, understanding the crest height variation of the surface waves remains a major challenge in offshore and nearshore industry due to limited measured time series surface elevation data.

Since the measured wave data are not available for large areas, satellite observations and wave hindcast models are used in deriving the wave parameters (Shanas et al., 2014). The numerical models have become a tool to get long-term wave information in high-resolution (spatial and temporal) in areas devoid of measurements (Appendini et al., 2014). The numerical models such as Wave Action Model (WAM) (WAMDI Group, 1988), WAVEWATCH III (WW3) (Tolman, 1991; 2014), Simulating WAves Nearshore (SWAN) (Booij et al., 1999), MIKE21 Spectral Waves (DHI, 2011) are commonly used (Chawla et al., 2007) and several wave hindcast studies were carried out in the Arabian Sea in the past (Remya et al., 2012; Sandhya et al., 2014; Samiksha et al., 2015; Amrutha et al., 2016). Recently, ocean state forecast has gained significance and has become a challenging task, considering the range of wide user community and varied demands (Amrutha et al., 2016). The numerical modelling studies in the Arabian Sea indicates that during the high sea states ($H_s > 4$ m), WAM underestimates the $H_s$ by 0.5 to 1 m (12 to 25%) (Samiksha et al., 2015). After the JONSWAP field experiment (Hasselmann et al., 1973) in the North Sea, many studies are conducted on the wave growth and decay in varying wind fields (e.g.: Young, 1999). Wave growth and decay characteristics in SWAN are investigated by Rogers et al. (2003). Because of the complex physics of the shallow water environment and the paucity of wind measurements at finer time scales of few minutes, the wave modelling in the shallow coastal waters remains a challenge (Aijaz et al., 2016). Previous studies in waves in the eastern AS have either not explicitly examined the characteristics of high waves, not had a data covering large spatial coverage nor focused on the short-term wave characteristics.

In this study, we examine the characteristics of high waves ($H_s > 3$ m) and the closeness of the estimate of wave height based on the numerical model during the growth and decay period based on the data collected at 9 to 15 m water





depth in the nearshore waters of the eastern AS. This article is outlined as follows: data and the methodology are presented in the next section. The results are presented in section 3, and the concluding remarks are given in the final section.

## 2 Data and methods

The domain of the present study is the eastern AS. The waves were measured simultaneously at four locations along the west coast of India covering a distance of 350 km (Fig. 1). At each measurement station, a 0.9 m Datawell Directional Waverider Mk III buoy (Datawell, 2009) was moored and the data collected at 1.28 Hz for the period from 5 June 2015 to 4 July 2015 is used in the study. The water depths at the measurement locations varied between 9 and 15 m (Table 1). Although these depths would vary during a tidal cycle, this variation of ~ 2 m is neglected in the present analysis. Heave is measured with 3% accuracy and with 1 cm resolution. The wave crest height ($H_c$) i.e. the height of the crest above mean water level and the wave trough height ($H_t$) i.e. the depth of trough below the mean water level for each individual wave are calculated from the heave time series data of 30-minutes duration. The wave height (H) of the individual wave is the sum of the crest height and trough height. The maximum wave height in a 30-minutes duration is $H_{max}$. The variance density spectrum of surface elevation (wave spectrum) is estimated from the measured buoy heave data of 30-minutes duration through a fast Fourier Transform. The spectral analysis results in wave spectrum with a resolution of 0.005 Hz from 0.025 Hz to 0.1 Hz and after that, it is 0.01 Hz up to 0.58 Hz (Datawell, 2009). The $H_s$, mean wave period ($T_{m02}$) and spectral peak period ($T_p$) are estimated from the variance density spectrum of surface elevation for data covering 30-minutes. From the half hourly wave spectra, the 90, 75, 50 and 25 percentile wave spectra are also computed by taking the respective variance density of each frequency bin for all high waves ($H_s > 3$ m). The different percentile lines (e.g. 90 percentile) indicate how often the spectral energy density is below a particular value (e.g. 90%) at a frequency. The wave direction is obtained following circular moments (Kuik et al., 1988). Wave and wind direction use the meteorological convention. The directional spectra are from the Maximum Entropy Method (Lygre and Krogstad, 1986). From the directional wave spectra, the percentage of surface variance density from different directions are estimated following equation (1).

$$m_0 = \int_{0.025}^{0.58} \int_a^b E(f,\theta)d\theta df \tag{1}$$

Where $E(f,\theta)$ is the directional spectral energy density, $d\theta$ is 4°, df is the resolution in frequency, a and b are 164° and 244° for waves from the south-southwest, 244° and 272° for waves from southwest-west and 272° and 348° for waves from the northwest. The other parameters evaluated are the maximum spectral energy density and spectral peakedness parameter ($Q_p$) (Goda, 1970). Measurements reported in this article are in Coordinated Universal Time (UTC) and the local time is 05:30 h ahead of UTC.



The third-generation spectral wave model WAVEWATCH III 4.18 (Tolman, 2014) is widely used at global and regional scales and its validity is extended to coastal areas with parameterizations of bottom dissipation and wave breaking. It is used for the wave hindcast during 5 June 2015 to 4 July 2015 (the period for which observations are available) at all the four study locations. The model is forced with ERA-Interim surface wind fields, produced by the European Center for Medium Weather Forecasts, decomposed to u and v components at 10-m height above the sea surface at every 6-h interval and a spatial resolution of $0.5^o$. The bathymetry for the offshore region is from ETOPO1, 1 Arc-Minute Global Relief Model (Amante and Eakins, 2009) and for the coastal region, the hydrographic chart issued by the Survey of India is used. For the large part of Indian Ocean domain and the Southern Ocean ($20^o$ E -$112^o$ E and $70^o$ S-$5^o$ N), the model grid resolution is $0.5^o$ x $0.5^o$ and is $0.1^o$ x $0.1^o$ for the North Indian Ocean ($65^o$ E-$90^o$ E and $5^o$ N - $25^o$ N). The resolution in wave direction is at $10^o$ and the wave frequencies are on a logarithmic scale from 0.04 to 0.5 Hz. The spatial distribution of the $H_s$ and the wind speed in the Indian Ocean is studied based on ERA-Interim reanalysis data with a spatial resolution of 0.5°. The reanalysis data used in this study are the zonal and meridional wind components at 10-m height and wave height at synoptic times (00:00, 06:00, 12:00, 18:00 UTC).

Many empirical expressions relating the $H_s$ and wind speed (U) are also proposed for different oceanic regions. In the present study, we examined the two recently proposed expressions i) Wang et al. (2017) and ii) Andreas and Wang (2007). Wang et al. (2017) observed that equation (2) could be used for estimating the $H_s$ from wind speed and is validated for the northwest coast of the United States.

$$H_s = 0.0143U^2 + 0.9626 \qquad (2)$$

Andreas and Wang (2007) proposed the equations (3) and (4) for estimating the $H_s$ from the wind speed in the coastal areas by considering the water depth (D).

$$H_s = 1.36\tanh\left[\frac{\ln(D/6)}{1.9}\right] \quad U \le 4 m/s \qquad (3)$$

$$H_s = 0.0134\tanh\left[\frac{\ln(D/9)}{1.3}\right]U^2 + 1.36\tanh\left[\frac{\ln(D/6)}{1.9}\right] - 16\left[0.0134\tanh\left[\frac{\ln(D/9)}{1.3}\right]\right] \quad U > 4 m/s \qquad (4)$$

## 3 Result and discussions

### 3.1 Bulk wave parameters

The $H_s$ varied from 0.7 to 5.5 m during the study period. The heights of waves during the monsoon often exceed 3 m. During 5 June to 4 July 2015, $H_s$ exceeded 3 m for 15 to 23% of the time at the four locations studied (Fig. 2). Similarly $H_s$ more than 4 m are present during 2.6 to 11.5% of the time at the four locations (Table 1). A total of 69-74% of the surface height variance in the study area is a result of swells (Fig. 2). During 14-21 June 2015, as the wind speed increased from 2 to 16 m/s, the wave height increased from 3 to 8.7 m (Fig. 3). The increase in the swell $H_s$ is from 1.7 to 4.5 m, while that for



the wind-sea is from 0.8 to 2.1 m. Although the relative changes in wave height become smaller when compared to the relative change in wind speed, the increase in wave height is in accordance with the increase in the wind speed. The high waves occurred almost at the same time at all the 4 locations as shown in Figure 4. It may be seen in the time series of the sea surface elevation data that there is a significant difference in what wave precedes a high wave. In the same time, the

maximum $H_s$ at the 2 locations which are at a water depth of 15 m is almost same ~5 m, at 13 m water depth, it is 4.9 m and at 9 m water depth, maximum $H_s$ is 4.2 m. The decrease in wave height with decreasing water depth observed may then be a consequence of energy dissipation when the waves feel the bottom (Komen et al., 1994). On 22 June 2015, a depression ARB 02 (IMD, 2015) formed 650 km northwest of Ratnagiri and caused high waves with $H_s$ up to 5.5 m on 23 June 2015 indicating that the synoptic scale weather evolution is another important factor for wave growth. The influence of this storm

is not significant at other 3 locations and hence $H_s$ more than 5 m is not observed at these locations on 23 June 2015. After the high wave height event, the $H_s$ decays and reach 3 m on 24 June 2015 0600 h. The value of the $H_{max}$ measured at different locations varied significantly (6.3 to 8.7 m) (Table 2).  Figure 5 shows the surface profile containing the extreme wave crests and extreme wave troughs measured at the study locations during the one-month period.

        The height of the top 20 waves of the 30-minutes record containing the highest $H_s$ at different locations indicates

that the second highest wave height is 10-20% less than the highest wave height (Fig. 6).  Amrutha and Sanil Kumar (2015) observed that the maximum wave height measured is ~ 8% lower than that obtained following the conventional Rayleigh distribution based on the data recorded at 13 m water depth. Forristall (1978) indicate that the Rayleigh distribution over-predicts the high wave by 7 to 8% of the measured value. The present study shows that the maximum wave height based on theoretical distribution is 2 to 5% lower than the measured value except at Honnavar, where it was 5% over that based on the

theoretical distribution (Fig. 7). As the wave height increases, wave period also increases to certain extent. Due to the increase in wave period, the wave length increases and wave start to feel the bottom and hence the height can change. Ratio of the water depth and wave length, d/L, is used to identify whether it is deep, intermediate or shallow water. When the $H_s$ is more than 3 m, the waves satisfy the intermediate wave regime (d/L is between 0.05 and 0.5). In intermediate water, waves will start to 'feel' the seabed and hence, their characteristics become modified by the bathymetry and hence highest waves

disagree with the theoretical values. Also, since the buoy is in the intermediate water depth, the measured waves are not representative of the large-scale wave field due to the disturbance by the local bathymetry. The scatter plot between the crest and the trough height for the waves with $H_s$ more than 3 m, shows an increased crest height and a reduced trough height (Fig. 8). The departures from the 1:1 slope is caused by wave nonlinearity (Whittaker et al., 2016). The $H_{max}$ is 1.31 to 2.57 times the $H_s$ and the most extreme wave height ($H_{max}$) is 1.50 to 1.62 times the $H_s$ for different locations (Table 2). For the Andrea

and New Year waves, the largest waves ($H_s$ 9.2 and 11.9 m respectively) recorded in the North Sea, the ratio of $H_{max}$ to $H_s$ is 2.3 and 2.1 (Bitner-Gregersen et al., 2014). In the present case, the crest heights of the individual wave in a 30-minute record vary from 0.75 to 1.70 times the $H_s$ of the same record.  In the same 30-minutes record, the most extreme crest height is 1.23 to 1.35 times the $H_s$ of the same 30-minutes data (Table 3). For the Andrea and New Year waves, the crest height is 1.63 and 1.55 times the $H_s$ (Bitner-Gregersen et al., 2014).  In the oceanographic community, when the crest height exceeds 1.25 $H_s$ or



when the crest-to-trough height is above twice the $H_s$, the waves are often referred to as "rogue" or "freak" waves (Kharif et al., 2009). Amrutha et al. (2014) and Sanil Kumar et al. (2013) observed that during the tropical cyclone, the average ratio of crest height to the corresponding wave height in 30 minutes record varied from 0.6 to 0.7. In the present study, the average ratio of crest height to the corresponding wave height is 0.58 to 0.59 at Karwar, Vengurla and Ratnagiri and is 0.67 for Honnavar. The large value at Honnavar is due to the shallowness of this location compared to other locations. The height of waves having maximum crest height in a 30-minutes record is sometimes smaller (lower by an average value of 4%) than the maximum wave height in the same record (Fig. 9). But the highest wave for the one-month period at each location is found to be the same having the highest crest height. Whereas the height of waves having the deepest trough height in a 30-minute record is much smaller than (up to 50%) the maximum wave height in the same record (Fig. 9). The recorded 30-minutes sea surface oscillations contain 190-270 waves during the high sea state. The most extreme trough height during the present study period is 0.85 to 1.01 times the $H_s$ (Table 3).

The wind forcing is one of the most important inputs to a wave model simulation. Generally, $H_s$ is proportional to the wind speed squared. $H_s$ are averaged in wind speed bins of 0.5 m/s width to study the correlation of $H_s$ with wind speed. All of the measured data points and the data averaged in wind speed bins are presented in Figure 10, together with the best fit to these averages and the Wang et al. (2017) and modified Andreas and Wang (2007) estimates. The study shows that the influence of sea surface wind speeds on wave height is significant during the monsoon. Even though the measured waves are predominantly swells, the wind–wave coupling is strong. But the good correlation between wind and high waves does not necessarily mean that these waves are directly caused by the wind around the measurement location. The principal focus of the present study is on the high waves and during the high waves, Andreas and Wang (2007) expression estimated the $H_s$ close to the measured value except at Vengurla.

During the one-month period (5 June to 4 July 2015), the high waves ($H_s > 3$ m) are from 240° to 270° (Table 4). The 20-m depth contours at the measurements locations are 15 to 30º inclined to the west from the north. The measured direction of the high waves indicates that they are aligned to the depth contours due to wave refraction. The waves of 3-4 m $H_s$ are with 10-15 s peak period (Fig. 11) and 6.5 to 8.6 s mean wave period (Table 4). The $H_s$ has better correlation with the mean wave period than with the peak wave period since $H_s$ and mean wave period has contributions from all spectral components, whereas the peak wave period has only a small subset. Since the majority of wave generation during the monsoon is within, or adjacent to, the study area, wave periods are rarely above 15 s. The average $T_p$ of high waves is around 12 s. The maximum wave heights are also not associated with the largest spectral peak period. The measured values of the $T_p$ corresponding to the maximum $H_s$ is 13.3 s. Alves (2006) and Amrutha et al. (2017) noted that the southern Indian Ocean westerly swells propagates eastward and reach the northern Indian Ocean. Although it is reported that longer-period swells ($T_p > 18$ s) generated in the southern Indian Ocean and the Southern Ocean are present in the eastern AS (Amrutha et al., 2017), during the high wave events, such long-period waves are absent.



### 3.2 Latitudinal variability in wave height

The latitudinal variability in wave parameters is studied based on ERA-Interim reanalysis data. Data along 70° E longitude from 60° S to 19° N latitude is considered in the study. To better understand the spatial variability of waves in the Indian Ocean during the high wave event, a Hovmöller diagram for significant wave heights is presented in Fig. 12. Hovmöller diagrams of $H_s$ shows that the $H_s$ is very high (~8-9 m) in the Southern Ocean and it is less (~1.5-2.5 m) in the equator region when 3-4 m $H_s$ is observed in the study area (Fig. 12). The wave period also shows a gradual decrease from the Southern Ocean to the study area. The wave direction in the Southern Ocean and the study area is southwest, whereas it is southeast at the equator (Fig. 12). Although there is evidence to suggest that waves in the eastern AS can be modified by strong wind events in the Southern Ocean. Fig. 12 indicate that the wave propagation from south Indian ocean/Southern Ocean to the study area is not significant during the peak monsoon period. Wind speed also shows a different pattern at the equator. The synoptic view of the wind speed along with $H_s$ in the Southern Ocean and the Indian Ocean on 21 June 2015 at 1200 h shows the large spatial variation in wind and wave parameter (Fig. 13). At the time of maximum $H_s$, the wind speed is strongest in the southwest. Figure 13 suggests that, the southerly-wind events are not directly associated with the strength of the waves at the eastern AS, which instead are associated with the strength of the winds in the western AS. The high waves in the study area are as a result of the strong cross-equatorial winds of the Findlater Jet (Findlater, 1969). High waves of 4 to 6 m occur in the western Arabian Sea and propagate to the study area. The separation of wave spectral energy from different directions indicate that for the 4 locations studied, ~57-63% of the surface variance density is from 244-272° (southwest-west) and ~20-27% are from 164-244° (south-southwest) and a small fraction is from the northwest (Fig. 14). As mentioned earlier, high waves at Ratnagiri on 23 June 2015 has larger northwest component since the energy provided by the storm added to the existing energy associated with the monsoon, whereas at other locations such difference is not observed. Two well-differentiated wave systems are present in the study area: a young swell (with peak wave periods below 12 s) from the southwest-west and a more developed swell (with peak wave periods over 12 s) from the south-southwest.

### 3.3 Wave spectra

It is also observed that all the wave systems are mono-modal and show similar spectral peak frequencies, mostly between 0.07 and 0.1 Hz (Fig. 15). The high spectral energy density is from 240-270° (Fig. 15). A perusal of the wave spectra during the wave growth and decay indicated differences from time to time. The wave spectra during 16 June 2015 1300 h to 24 June 2015 1300 h show that the peak frequency shifts from higher (0.09 Hz) to lower (0.075 Hz) during the wave growth and from lower (0.075 Hz) to higher (0.08 Hz) during the decay (Fig. 16). During the decay (21 June 2015 to 24 June 2015), at higher frequencies, the decay of wave spectral energy density is less compared to that at frequencies just above the peak frequency. The spectral peakedness parameter exhibit a relatively high scatter (1.5 to 3.3) for $H_s$ more than 3 m with an average value of 2 to 2.2 for different locations. The wave spectrum for the highest waves has peakedness parameter ranging from 2.4 to 2.9 at different locations. To understand the spatial variation in the spectral energy density of



high waves, the 90, 75, 50 and 25 percentile wave spectra at 4 locations are estimated as in section 2 and presented in Figure 17. There is no significant spatial variation in the spectral energy at the three locations (which are at 13 to 15 m water depth) excluding Honnavar, except the spectral energy density at the peak frequency. All the wave spectra seen in Figure 17 indicate lower spectral energy values at Honnavar, since it is at a relatively shallower depth (~ 9-m).

**3.4    Numerical model results**

In order to assure WAVEWATCH III wave hindcast data reliability during the high wave events, we compared the wave hindcast $H_s$ data and the measured wave data during 5 June 2015 to 4 July 2015. The bias, correlation coefficient (r), and scatter index (SI) is estimated to provide an indication of the wave model performance during the wave growth and wave decay. Figure 18 shows the scatter plot between the measured and hindcast $H_s$ data. The dots represent the data and the line

is the exact match line. The $H_s$ model results are reasonable and follow the measurements very well during the wave growth period. The coefficients of correlation between the model and the measured data are ~0.97. The error statistics showed that at all the four locations, $H_s$ had a good match with measured values (Fig. 18). The comparison of hindcast data with measured data shows a slight under-estimation (negative bias 0.05 to 0.24 m) of the $H_s$ during the wave growth and a large over-estimation (positive bias 0.47 to 0.65 m) during the wave decay (Fig. 18).

**4 Concluding remarks**

In this study, an analysis of the high waves during the southwest monsoon from 5 June 2015 to 4 July 2015 is presented. The work is based on measured wave data from four nearshore buoys in intermediate waters using the directional waverider buoys. The buoy data indicate south-westerly waves of 3-4 m significant wave height and 10–15 s peak period.

The average ratio of crest height to the corresponding wave height is 0.58 to 0.67 and the higher value is observed at the shallow location. The wave propagation from south Indian Ocean/Southern Ocean to the study area is not significant during the peak southwest monsoon period. The largest wave heights are associated with the southwest wind events. The comparison of simulations run with WAVEWATCH III with measured data shows a slight under-estimation of the significant wave height during the wave growth and a significant overestimation during the wave decay. The results

presented in this paper could benefit not only the scientific community but also the ocean engineering community for the design of marine structures.

**Acknowledgments**

The authors acknowledge the Director, CSIR-NIO for providing the facilities to conduct this research and the Earth System

Science Organization-Ministry of Earth Sciences, New Delhi for providing the financial support to conduct this research. We thank TM. Balakrishnan Nair, Head OSISG, Arun Nherakkol, Scientist, INCOIS, Hyderabad, RS. Kankara, Head, Coastal Processes & Shoreline Management, ICMAM-PD, Chennai and Jai Singh, Technical Officer, CSIR-NIO for help during data collection. The authors acknowledge the CSIR-NIO high-performance computing facilities made available for conducting



the research. We thank Vedpathak and Parag Kulkarni, Centre for Coastal & Marine Biodiversity, Dr. Babasaheb Ambedkar Marathwada University, Ratnagiri and JL Rathod, Department of Marine Biology, Karnataka University PG Centre, Karwar for providing the logistics required for wave data collection at Ratnagiri and Karwar. We thank the topical editor John M. Huthnance for the suggestions to improve the readability of the manuscript. This publication is a NIO contribution.

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

Table 1. Measurement locations and the water depth

| Station | Latitude Longitude | Water depth (m) | Percentage of time | |
|---|---|---|---|---|
| | | | $H_s > 3$ m | $H_s > 4$ m |
| Ratnagiri | 16.980˚ N; 73.258˚ E | 13 | 22.6 | 11.5 |
| Vengurla | 15.842˚ N; 73.563˚ E | 15 | 17.7 | 8.1 |
| Karwar | 14.821˚ N; 74.053˚ E | 15 | 19.6 | 8.3 |
| Honnavar | 14.304˚ N; 74.391˚ E | 9 | 15.4 | 2.6 |

Table 2. Range and average (Avr) value of various wave height parameters along with the crest and trough height ratios for records with maximum wave height equal or above 3 m

| | Ratnagiri | | Vengurla | | Karwar | | Honnavar | |
|---|---|---|---|---|---|---|---|---|
| | Range | Avr | Range | Avr | Range | Avr | Range | Avr |
| $H_s$ (m) | 1.38-5.48 | 2.74 | 1.30-5.05 | 2.68 | 1.25-5.02 | 2.62 | 1.42-4.20 | 2.46 |
| $H_c$ (m) | 1.43-5.87 | 2.75 | 1.41-5.41 | 2.72 | 1.41-5.56 | 2.65 | 1.57-5.44 | 2.64 |
| $H_t$ (m) | 1.30-3.92 | 2.23 | 1.28-3.95 | 2.18 | 1.26-3.99 | 2.15 | 1.19-3.31 | 1.78 |
| $H_{avg}$ (m) | 0.86-3.11 | 1.70 | 0.82-3.01 | 1.67 | 0.80-2.88 | 1.63 | 0.91-2.66 | 1.55 |
| $H_{max}$ (m) | 3.00-8.69 | 4.56 | 3.00-8.19 | 4.49 | 3.00-7.57 | 4.38 | 3.00-6.28 | 4.04 |
| $H_c/H_s$ | 0.75-1.49 | 1.00 | 0.80-1.48 | 1.02 | 0.77-1.70 | 1.01 | 0.83-1.63 | 1.07 |
| $H_t/H_s$ | 0.63-1.15 | 0.83 | 0.63-1.27 | 0.83 | 0.64-1.24 | 0.83 | 0.56-1.14 | 0.74 |
| $H_{max}/H_s$ | 1.33-2.29 | 1.68 | 1.31-2.41 | 1.70 | 1.37-2.57 | 1.68 | 1.32-2.24 | 1.66 |
| $H_c/H_{max}$ | 0.46-0.75 | 0.60 | 0.42-0.77 | 0.60 | 0.45-0.76 | 0.60 | 0.48-0.78 | 0.65 |
| $H_t/H_{max}$ | 0.35-0.67 | 0.49 | 0.35-0.68 | 0.49 | 0.33-0.70 | 0.50 | 0.33-0.64 | 0.45 |





Table 3. The ratio of crest height to wave height and trough height to wave height in a 30-minutes record during the occurrence of maximum crest and maximum trough heights

| | Ratnagiri | Vengurla | Karwar | Honnavar |
|---|---|---|---|---|
| Parameters | | Crest maximum | | |
| $H_c$ (m) | 5.87 | 5.41 | 5.56 | 5.15 |
| $H_{max}$ (m) | 8.60 | 8.40 | 7.57 | 7.01 |
| $H_s$ (m) | 4.36 | 3.85 | 4.51 | 3.83 |
| $H_c/H_s$ | 1.35 | 1.41 | 1.23 | 1.34 |
| $H_c/H_{max}$ | 0.68 | 0.64 | 0.73 | 0.73 |
| | | Trough maximum | | |
| $H_t$ (m) | 4.37 | 3.95 | 3.99 | 3.28 |
| $H_{max}$ (m) | 6.57 | 7.72 | 6.89 | 7.01 |
| $H_s$ (m) | 4.43 | 3.89 | 4.29 | 3.83 |
| $H_t/Hs$ | 0.88 | 1.01 | 0.93 | 0.85 |
| $H_t/H_{max}$ | 0.60 | 0.51 | 0.58 | 0.40 |

Table 4. Range and average (Avr) value of various wave parameters for records with significant wave height equal or above 3 m

| | Ratnagiri | | Vengurla | | Karwar | | Honnavar | |
|---|---|---|---|---|---|---|---|---|
| | Range | Avr | Range | Avr | Range | Avr | Range | Avr |
| $H_s$ (m) | 3.00-5.48 | 3.92 | 3.01-5.05 | 3.92 | 3.01-5.02 | 3.85 | 3.00-4.20 | 3.56 |
| $H_{max}$ (m) | 4.04-8.69 | 5.98 | 4.11-8.19 | 5.92 | 4.15-7.57 | 5.88 | 3.92-6.28 | 5.24 |
| $T_{m02}$ (s) | 6.4-8.6 | 7.5 | 6.4-8.7 | 7.6 | 6.6-8.5 | 7.4 | 6.7-8.1 | 7.4 |
| $T_p$ (s) | 9.1-14.3 | 12.3 | 10.0-14.3 | 12.4 | 10.0-14.3 | 12.4 | 10.0-14.3 | 12.5 |
| $T_{Hmax}$ (s) | 6.9-13.7 | 10.5 | 7.7-13.6 | 10.7 | 7.1-13.7 | 10.6 | 7.9-15.8 | 10.9 |
| Dir (deg) | 248-270 | 258 | 243-263 | 254 | 245-267 | 255 | 246-269 | 258 |





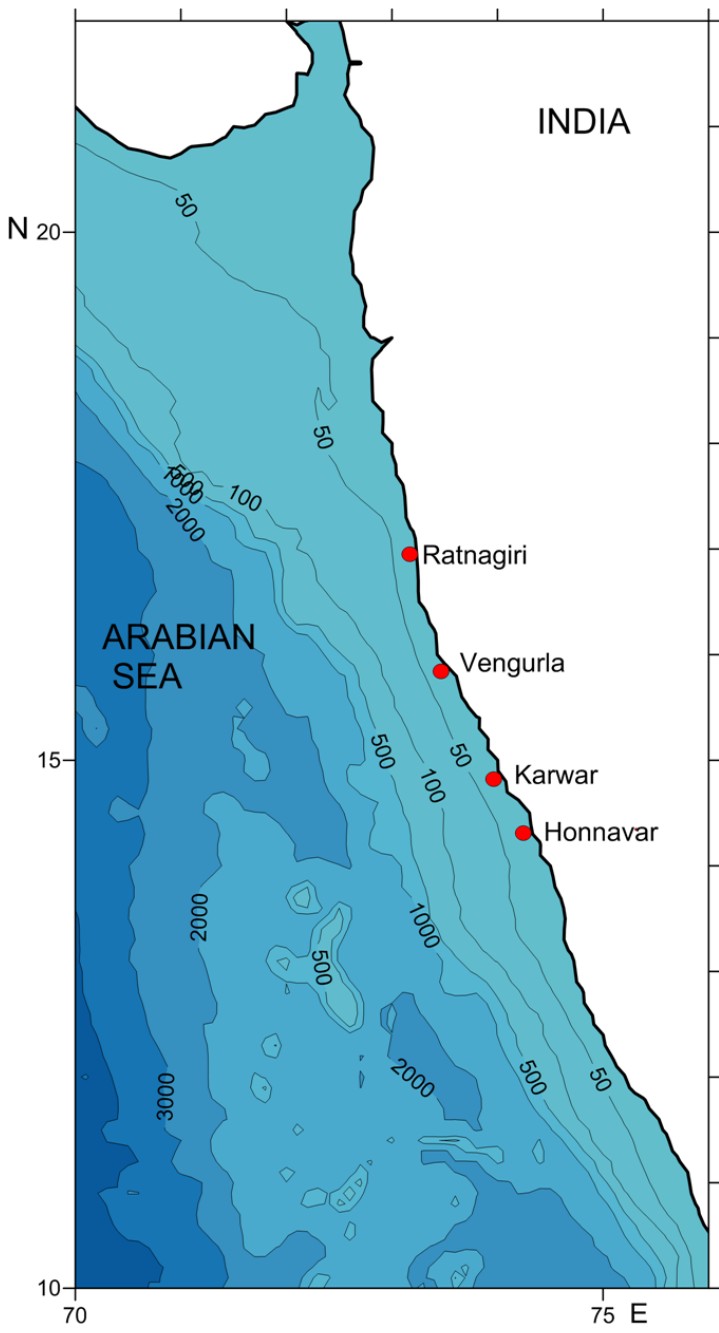

Figure 1. Location map of four directional wave measurement locations in eastern Arabian Sea





Figure 2. Time series plot of significant wave height measured at four locations from 5 June to 4 July 2015 History of significant wave height for wind-sea and swell are also presented.





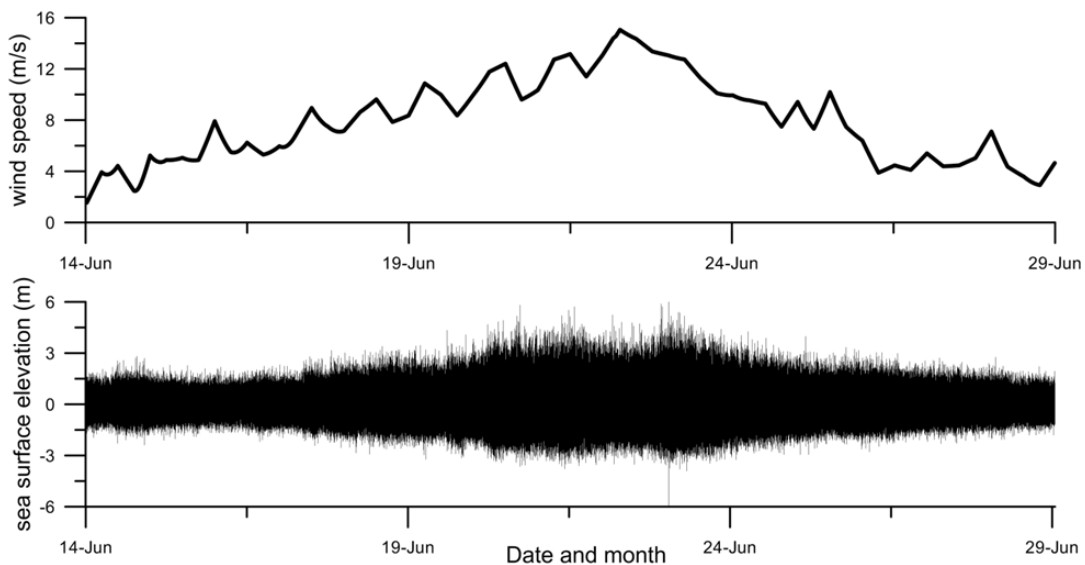

Figure 3. Time series plot of wind speed and the sea surface elevation during 14-28 June 2015 at Ratnagiri

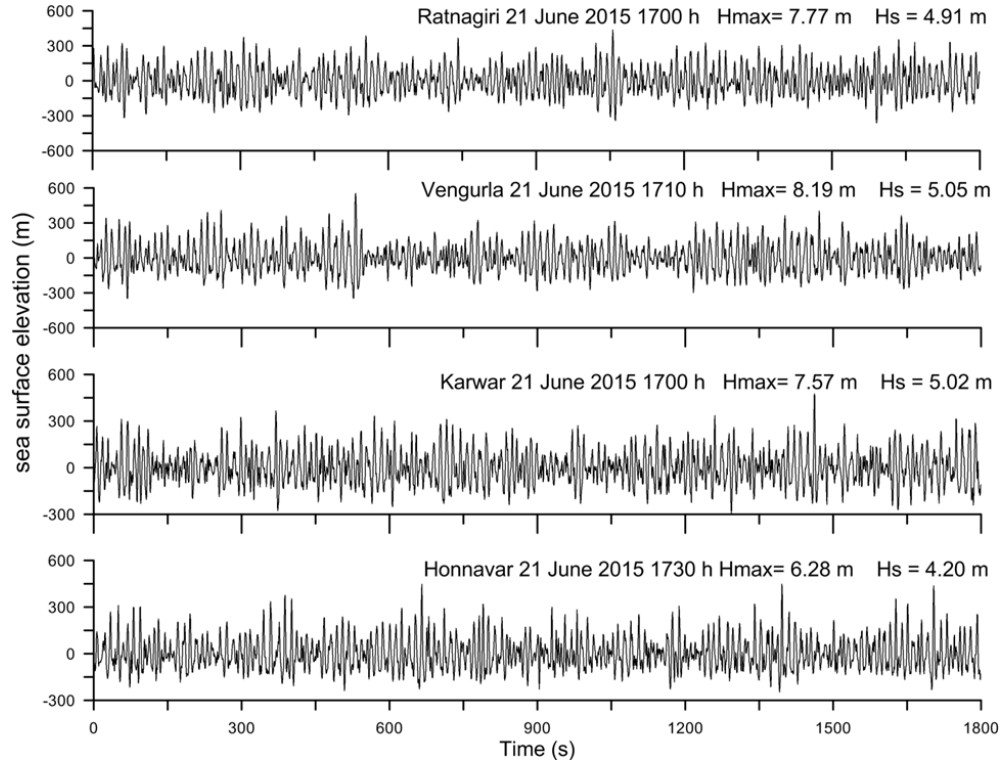

Figure 4. Surface elevation time series record of 30-minutes duration containing the highest wave at different locations





Figure 5. Surface elevation time series record containing extreme wave crest and trough measured at different locations





Figure 6. a) Height of the top 20 waves in the 30-minutes record containing the highest significant wave height, b) ratio of the crest height and wave height, c) ratio of crest height and trough height for all the 4 locations studied





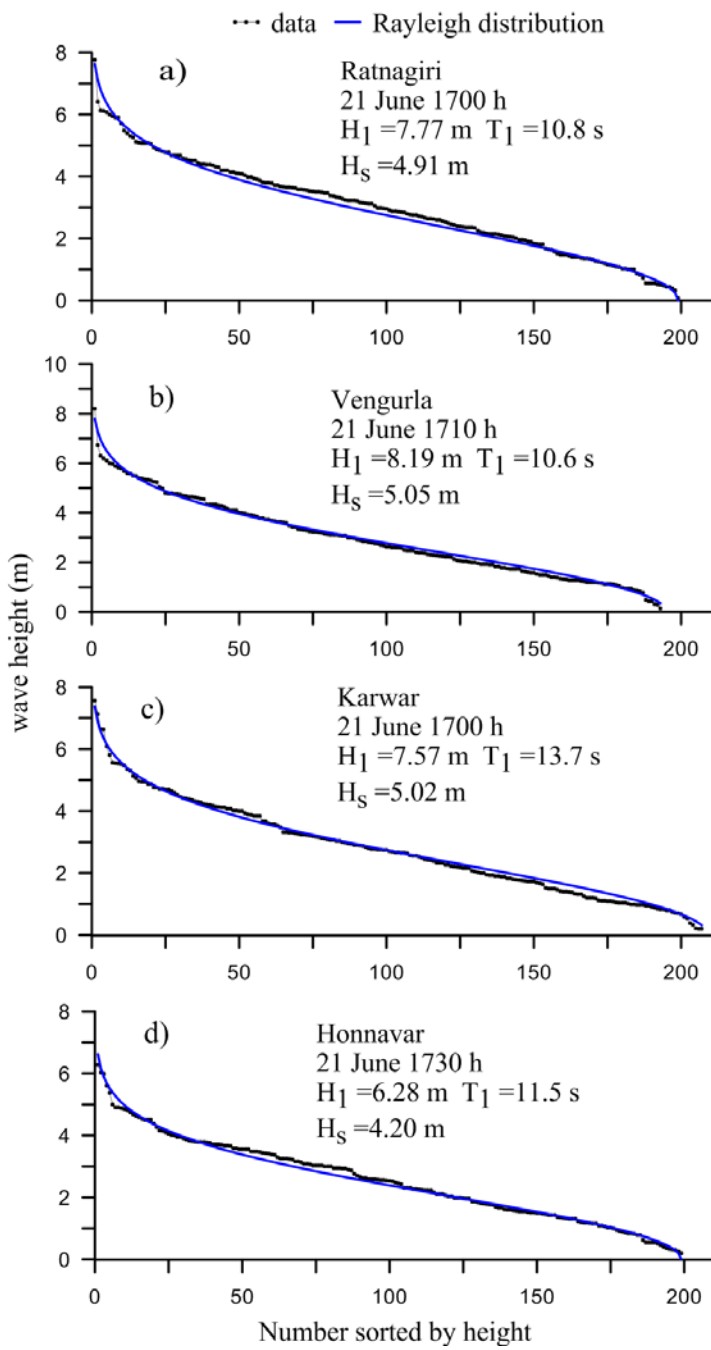

Figure 7. Plot of wave height measured and estimated following Rayleigh distribution during 30-minutes for selected time at different locations






Figure 8. Scatter plot of crest and trough heights from 30-minutes record for waves with significant wave height more than 3 m, showing departures from the 1:1 slope. The grey dots indicate all the data and the blue cross is for maximum wave height data in the 30-minutes record



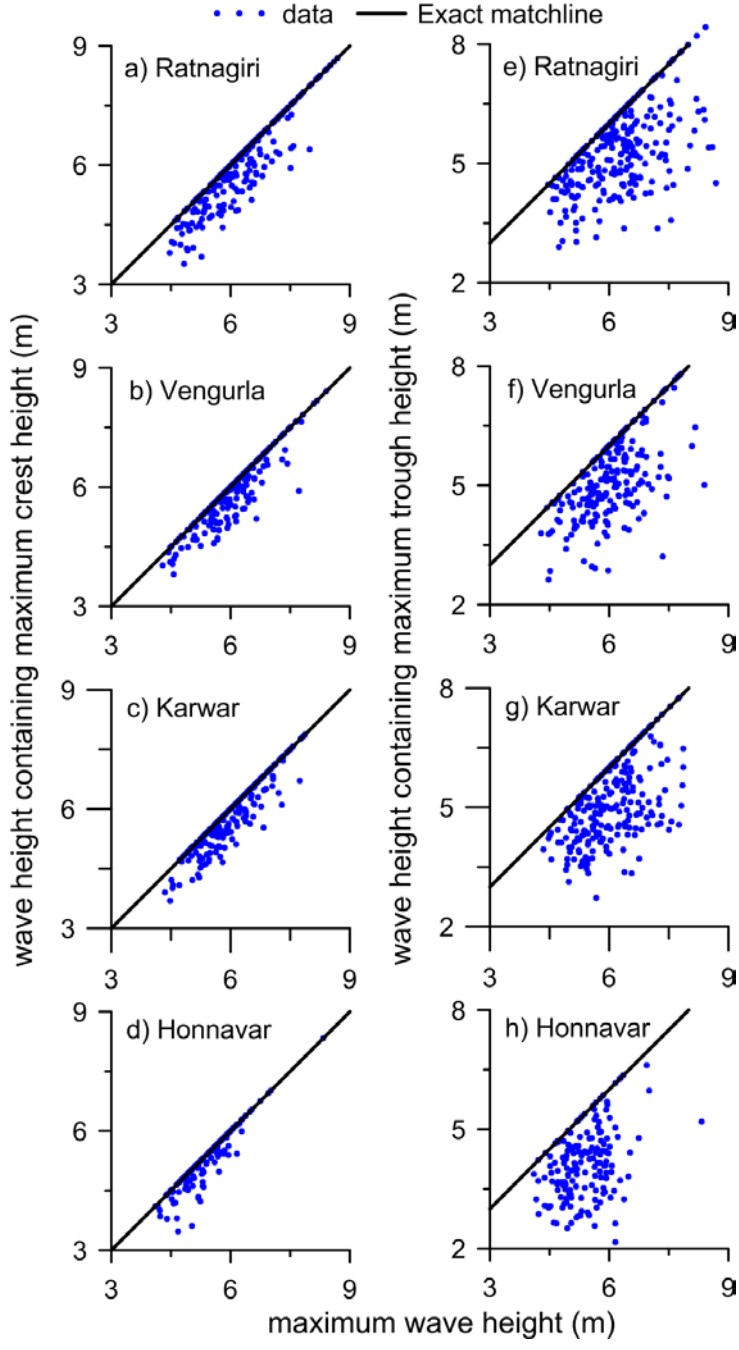

Figure 9. Left panel shows the scatter plot of wave height containing maximum crest height with maximum wave height in 30-minutes record and the right panel is between the wave height containing deepest trough height and maximum wave height in 30-minutes record



Figure. 10. Significant wave heights versus 10-m wind speeds (gray circles) at Ratnagiri, Vengurla, Karwar and Honnavar.

Blue crosses are significant wave heights averaged in wind speed bins that are 0.5 m s$^{-1}$ wide. The corresponding curves

5    (solid line) are the best fit to these averages





Figure 11. Plot of significant wave height with mean wave direction and peak wave period for the locations studied. The wave parameters over the half-hour record segments are presented for one month



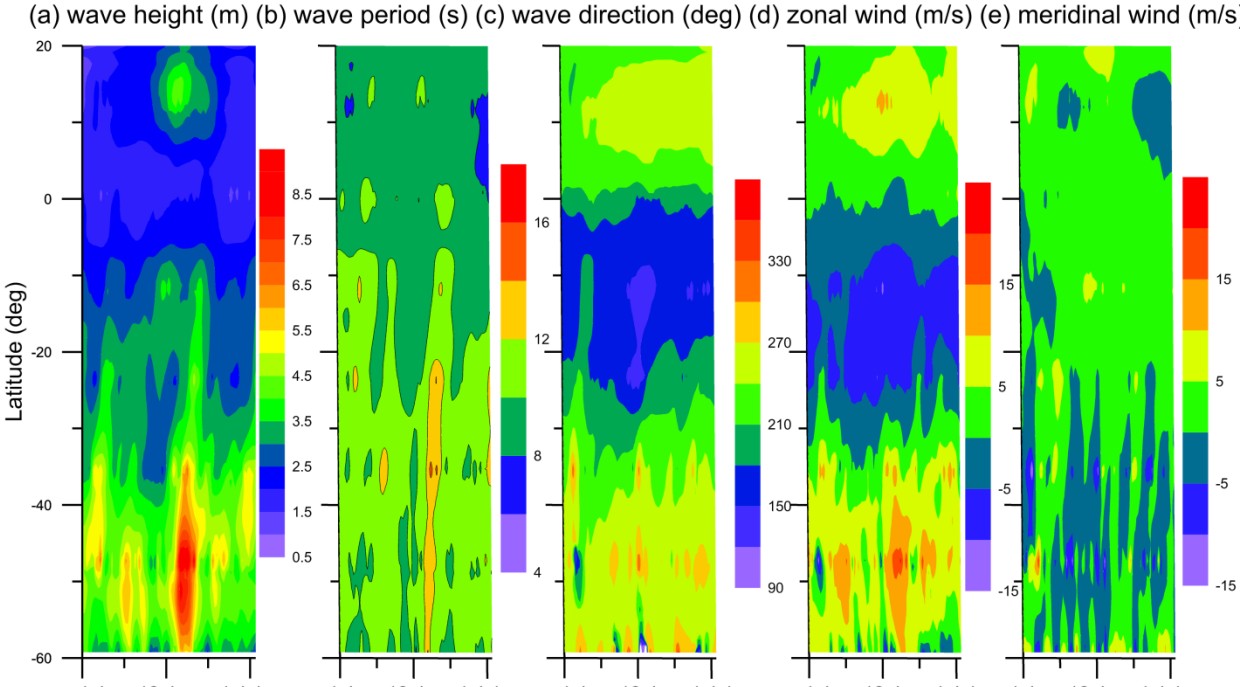

Figure 12. Hovmöller diagrams of a) significant wave height, b) wave period, c) wave direction, d) zonal and e) meridional wind speed showing the latitudinal variability in wave parameters along 70° E longitude from 60° S to 20° N latitude. The

5    time    is    shown    along    the    x    axis    and    the    latitude    of    the    wave    is    shown    along    the    y    axis





Figure 13. Synoptic view of the wind speed (left panel) and significant wave height (right panel) in the Southern Ocean and

5    Indian Ocean on 19 June 2015 at 1200 h (top) and 21 June 2015 at 1200 h (bottom)





Figure 14. Percentage of surface height variance from different directions (0-164° & 348-360°, 164-244°, 244-272° and 272-348°) at the locations studied





Figure 15. Top panel shows the temporal variation of spectral energy density ($m^2$/Hz) with frequency at four locations during 5 June to 4 July 2015. Bottom panel shows the temporal variation of spectral energy density ($m^2$/Deg) with direction.

5   Spectral energy density is shown in logarithmic scale (base 10)



Figure 16. Wave spectra in different days during the wave growth (16, 18, 19, 20 and 21 June 2015) and wave decay (23 June 2015)





Figure 17. Percentiles ( 90, 75, 50 and 25) of wave spectra for the 4 locations studied for all the high wave events ($H_s > 3$m). The significant wave height estimated from the resultant spectrum is also presented





Figure 18. Scatter plot of significant wave height estimated from model with the measured ones. Left panel shows the wave height during the growth stage and right panel is that during the decay. Straight line is the exact match line