# Peer review of "Characteristics of high monsoon wind-waves observed at multiple stations in the eastern Arabian Sea"

_Ocean Science, 2017_

## Referee Comment (RC1) · Anonymous Referee #1 · 13 Dec 2017

Review of Manuscript os-2017-84 "Characteristics of high monsoon wind-waves observed at multiple stations in the eastern Arabian Sea" by Amrutha and Kumar"

Recommendation: Reject

Fundaments: I am afraid that based on only 1 month wave data and reanalysis data, it is not feasible to characterize the impact of the monsoon on the wave field in the area. Then this would be a case study, which does not seem to be the case. I am afraid the study has way too many flaws and inaccuracies to be accepted as a publication.

Several parts of the methodology are not well explained on introduced into the research context.

[Figure]

Several parts of the text are confusing and meaningless.

Additional comments:

Abstract: Confusing. Not really an abstract from the academic requirements P1L7: Why is this specific month a typical Indian summer monsoon month? P1L16: Numerical wave model? Maybe reanalysis. P1L25: What are monsoon waves? P1L6: Why bringing the Rayleigh distribution into the discussion (introduction)? P2L19: WAM does not mean wave action model. P3L10: waves are surface perturbations of the still water level, not the mean water level (those are tides). P3L17: What are percentile wave spectra? P3L19: Below or above? P4L6/L12: The ERA-Interim resolution is not 0.5 dgrs. P4L28: How do u separate wind sea and swell? How do you know this is just swell? P5L25: What are theoretical values? P5L29/30: What are Andrea and New Year waves? P6L15/L16: "The study shows that the influence of sea surface wind speeds on wave heights is significant during the monsoon". And we didn't know that before? And do you really show that by using just 1 month? P6L17: "... Predominantly swells"? How do you know? Speculative. P7L16: The wind field in the area is way richer than that, and the Findlater jet is just part of it. Please investigate the role of the Oman coastal low level jet.

---

## Referee Comment (RC2) · Anonymous Referee #2 · 18 Dec 2017

The paper analyses wave measurements for one month at four locations in the Arabian Sea. From scattered sentences in the introduction and concluding remarks, I can understand the objective of this paper might be to study the occurrence of high waves in the Arabian Sea and its implications for the ocean engineering community, but this is not really what is presented in the paper. Methodology and results are not clear, nor the part related to observations and even less the wave modelling part. The paper does not hold together. Wave observations cover a too short time period, the wave modelling part of the study is a merely one month validation. I do not see any scientific conclusions that can be drawn from the presented data and methods. I recommend to reject it.

**Abstract**

- One-month period is too short for drawing any conclusions.

- How do we know this is a typical Indian summer monsoon?

- I would expect to have in the abstract the main objectives and the significant results. In the present form, I do not understand the significance on the numbers given here.

**1 Introduction**

I would expect in the Introduction to have a list of previous studies addressing the scientific issue presented in this paper and a statement clarifying how the authors are advancing in the field. The list of wave models with their references is meaningless for this study, it is not the state-of-the-art this paper would need (and you do not discuss why you chose one model instead of another one).

**2 Data and methods**

page 4 line 1-13: The need of using a wave numerical model in this work is obscure.

page 4 line 14-22: It is not clear the aim for presenting the empirical relations between Hs and wind speed here.

**3 Results**

Results are hard to read and it is difficult to capture the most relevant points.

page 4 line 27: You did not explain how you separated swell and wind-sea waves. In Fig. 2 it is not clear where is "History of significant wave height for wind-sea and swell are also present" (as written in the label).

page 5 line 4: This sentence is not clear.

page 5 line 7-13: During the one-month period of data, there is a storm and the Hs maximum values observed are recorded during this storm. Are such storms typical during the monsoon season? What's the return period/probability of occurrence of this Hs value in the region? I found these numbers meaningless without a context.

page 5 line 29 to page 6 line 11: The focus is now on freak waves. Am I right in understanding that no freak waves were recorded during the one-month period of time? Has this any significance?

page 6 line 12: Why talking about wind forcing in wave model simulations here?

page 6 line17: It is not clear how the authors know the measured waves are mainly swell. It not clear the meaning of wind-wave coupling and how the reader can understand it is strong.

page 8 line 6-14: You have set up a wave model for the area of interest and compared with observation for a very limited time period. Nowadays, this is not worth publications in any scientific journal. It would have made sense to present the wave model setup and its validation, if then you were going to use it for a specific purpose. Numerical models are helpful to complement observations, for example if observations are scattered in space or limited in time. It is missing what the authors want to study with the model they have validated. Extending the time period and the spatial domain of observations could be an idea.

**4   Conclusions**

It is given as a statement, but it is not clear how the scientific and ocean engineering communities can benefit from the results presented in this paper.

---

## Referee Comment (RC3) · Anonymous Referee #3 · 18 Dec 2017

Scientific significance: Does the manuscript represent a substantial contribution to scientific progress within the scope of Ocean Science (substantial new concepts, ideas, methods, or data)?

Fair. I do not find any novel concepts, ideas or methods. Previous studies have detailed many aspects of wave climatology and characteristics at the same locations. Here, the authors seem to focus on crest/trough ratios for the upper percentile of observed wave height as their novel contribution, along with a rough quantification of the accuracy of a numerical model to capture wave growth/decay dynamics (model validation), thereby, the data may represent a contribution pertinent to Ocean Science.

Scientific quality: Are the scientific approach and applied methods valid? Are the results discussed in an appropriate and balanced way (consideration of related work, including appropriate references)?

Good. The applied methods are reasonable.

Presentation quality: Are the scientific results and conclusions presented in a clear, concise, and well-structured way (number and quality of figures/tables, appropriate use of English language)?

Poor. The presentation is not cohesive, is overly complex, and at times irrelevant in its attempt at comprehensiveness. It includes many well-known, one might say, gratuitous statements as apparent motivations or conclusions. If I am correct that the crest/trough ratios are the primary "characteristic" addressed, this is not clearly communicated in the Abstract or Introduction. The other primary goal to assess wave model growth/decay is not clearly identified until the results are presented.

Decision: The manuscript is not acceptable for publication in Ocean Science. A major revision of the manuscript is required to bring it to acceptable standards.

Comments:

1) This paper does not put forth a hypothesis, it is a data analysis and model validation presentation. In this case, the novel data and it's novel analysis should be highlighted. The Abstract spends much effort to describe the data, but does not communicate clearly that the crest/trough of extreme waves and numerical growth/decay accuracy are the contributions of this work.

2) The Introduction provides a fragmented description of the problem background and motivation. For example, pg 1 line 23 and pg 2 line 4 both cite studies that have studied the "characteristics" of monsoon waves, but no clarification is given as how, or why this study is differentiated from those.

3) Page 2 lines 8-15 are apparently the place where the primary goal/contribution of

this paper is expressed, but it is not clear or concise.

4) Page 2 lines 16-31. This paragraph is confusing and fragmented at-best. It starts with a review of contemporary model studies, with no clear justification as to why they are mentioned, then confusingly refers to JONSWAP as a motivation for wave growth/decay studies. It then suggests that shallow water environment is complex, and then suggests that the wave "characteristics" of "high waves" in shallow water have not been examined. Aside from being an untrue assertion, this paragraph only deals in the abstract generality of wave "characteristics".

5) Page 2 lines 32-33 Page 3 lines 1-2. The first real expression of the intended goals of this paper. Please clarify "characteristics".

6) Given all the above, my suggestion is to completely rewrite the Introduction, in a focused and concise manner clearly communicating the novel aspects of this study, without the need to cite work that is rather irrelevant.

7) The Data and Methods sections do not describe how the "closeness of estimates" is determined.

8) Page 4 lines 16-18. Why would an empirical model validated on the Northwest coast of North America in the north Pacific apply to coastal conditions in the Arabian Sea?

9) Page 4 lines 28-29. It is not detailed how swell and wind waves are differentiated.

10) ibid. Is the wind speed observed or modeled? At what location? What is the integration period?

11) Page 5 "Although the relative changes in wave height become smaller when compared to the relative change in wind speed, the increase in wave height is in accordance with the increase in the wind speed." – This statement is ambiguous and without clarification, gratuitous.

12) Page 5 "It may be seen in the time series of the sea surface elevation data that

there is a significant difference in what wave precedes a high wave." – Ambiguous

13) Page 5 "indicating that the synoptic scale weather evolution is another important factor for wave growth." – Gratuitous

14) Page 5 line 17. Details on the Rayleigh distribution parameters and fits are needed.

15) Page 5 lines 21-24. This explanation for "theoretical disagreement" is very simplistic and unsatisfying. Why can't a more precise, quantifiable argument be made?

16) Page 5 line 25. "since the buoy is in the intermediate water depth, the measured waves are not representative of the large-scale wave field due to the disturbance by the local bathymetry." – I would argue that the observed waves are representative of the large-scale wave field. That the observed intermediate/shallow water waves are modulated in relation to the deepwater waves is expected.

17) Page 5 line 28. "The departures from the 1:1 slope is caused by wave nonlinearity"

18) Page 5 line 26 - page 6 line 11, Figures 8 and 9. This material should be in a standalone subsection. Is it not one of the two novel contributions of this work?

19) Page 5 line 26 - page 6 line 11, Figures 8 and 9. If the authors desire to increase the relevance of the paper, a connection to the underlying physics of the bathymetry and wave response to this data would be a step in that direction.

20) Page 6 line 12. "The wind forcing is one of the most important inputs to a wave model simulation." – Gratuitous

21) Page 6 line 14 Figure 10. There is no explanation of what a "best fit" is, or how it is determined. There is no metric quantifying the veracity or goodness-of-fit of the different models.

22) Page 6 line 15. "The study shows that the influence of sea surface wind speeds on wave height is significant during the monsoon." – Gratuitous

23) Page 6 line 18 "The principal focus of the present study is on the high waves and during the high waves" Why is this here? It should be in the Introduction.

24) Page 7 line 25 "A perusal of the wave spectra during the wave growth and decay indicated differences from time to time." – Gratuitous

25) Page 8 lines 10-14. There should be standard errors or confidence bounds on all the presented statistics.

26) Figure 1. There are no units.

27) Figure 2. The vertical dashed lines are not denoted.

28) Figure 3. Details of the wind speed are needed. I don't see the need for the lower panel of water level variance. Are not the plots in figure 2, plots of water level variance? I would find it more cohesive to remove the variance in this figure and move the wind speed to figure 2.

29) Figure 4. The ordinal units are probably wrong. There is no datum specified for the water levels. It would be better if all 4 records covered the same time period.

30) Figure 5. There is no datum specified for the water levels.

31) Figure 6. I don't see the need to include the plots of crest/wave and crest/trough unless you are going to analyze the data in these plots (via regression or some model). There isn't much information that the reader can gain from looking at these.

32) Figure 10. As mentioned above, there is no metric of fit veracity, or description of how the "best fit" was computed.